# Comparing the reliability of different ICA algorithms for fMRI analysis

**Pengxu Wei**[1,2], **Ruixue Bao**[3], **Yubo Fan**[2]*

**1** Beijing Key Laboratory of Rehabilitation Technical Aids for Old-Age Disability, Key Laboratory of Neuro-Functional Information and Rehabilitation Engineering of the Ministry of Civil Affairs, National Research Center for Rehabilitation Technical Aids, Beijing, China, **2** Key Laboratory of Biomechanics and Mechanobiology (Beihang University), Ministry of Education, Beijing Advanced Innovation Center for Biomedical Engineering, School of Biological Science and Medical Engineering and Beijing Advanced Innovation Center for Biomedical Engineering, Beihang University, Beijing, China, **3** School of Rehabilitation Medicine, China Rehabilitation Research Center, Capital Medical University, Beijing, China

* yubofan@buaa.edu.cn

**Data Availability Statement:** All relevant data are within the manuscript and its Supporting Information files.

**Funding:** This research was funded by the National Key R&D Program of China (Grant Nos. 2018YFC2001400 and 2018YFC2001700 by WP

## Abstract

Independent component analysis (ICA) has been shown to be a powerful blind source separation technique for analyzing functional magnetic resonance imaging (fMRI) data sets. ICA can extract independent spatial maps and their corresponding time courses from fMRI data without a priori specification of time courses. Some popular ICA algorithms such as Infomax or FastICA generate different results after repeated analysis from the same data volume, which is generally acknowledged as a drawback for ICA approaches. The reliability of some ICA algorithms has been explored by methods such as ICASSO and RAICAR (ranking and averaging independent component analysis by reproducibility). However, the exact algorithmic reliability of different ICA algorithms has not been examined and compared with each other. Here, the quality index generated with ICASSO and spatial correlation coefficients were used to examine the reliability of different ICA algorithms. The results demonstrated that Infomax running 10 times with ICASSO could generate consistent independent components from fMRI data sets.

## Introduction

Independent component analysis (ICA) has been shown to be a powerful blind source separation technique for analyzing functional magnetic resonance imaging (fMRI). ICA can decompose a set of signal mixtures into a corresponding set of statistically independent component signals (source signals) [1–4]. The basic concept of ICA can be expressed as $X = M \cdot C$, where X is the observed data (i.e., the data matrix of fMRI signal), C is the component map (a matrix of voxel values), and M is the mixing matrix determining the time-varying contribution of each component map to the observed fMRI data. To estimate M and C simultaneously, ICA determines the unmixing matrix W (a permuted version of the inverse of the mixing matrix M) through iterative calculation. The component maps and corresponding time courses can be acquired using the following equation: $C = W \cdot X$ [5, 6].

and ZL), the National Natural Science Foundation of China (Grant No. 81972160 by WP), and the Beijing Natural Science Foundation (Grant No. 17L20019 by WP). The funders had no role in study design, data collection and analysis, decision to publish, or preparation of the manuscript.

**Competing interests:** The authors have declared that no competing interests exist.

A number of ICA algorithms have been used for fMRI data analysis, including Infomax (Information Maximization), FastICA, ERICA (Equivalent Robust Independent Component Analysis), and SIMBEC (SIMultaneous Blind signal Extraction using Cumulants). ICA can extract independent spatial maps and their corresponding time courses from fMRI data without a priori specification of time courses. However, most ICA algorithms are based on minimization or maximization of an objective function and will find different local minima depending on where the "initial point" is. The randomness of the initialization that inherently exists in such ICA algorithms introduces randomness into the ICA decomposition and leads to different results when running ICA repeatedly. As a result, some popular ICA algorithms such as Infomax or FastICA will generate different results after repeated analysis from the same data volume, which is generally acknowledged as a drawback for ICA approaches [7, 8].

The reliability of some ICA algorithms has been explored by methods such as RAICAR (Ranking and Averaging Independent Component Analysis by Reproducibility) [7] and ICASSO (http://research.ics.aalto.fi/ica/icasso/) [8]. Such methods repeat the analysis multiple times with different random initializations and subsequently quantify the consistency of the outcomes [7–13]. ICASSO provides several indices including the quality index (Iq) to assess the reliability of independent components (ICs) after multiple runs. The Iq is introduced to be the main index reflecting the compactness and isolation of a cluster that contains ICs with high similarities from multiple runs. The Iq is computed as the difference between the average intra-cluster similarities and average extra-cluster similarities. RAICAR uses the spatial correlation coefficients (SCCs) to measure the component reproducibility. The reliability of the FastICA algorithm was assessed by using RAICAR or its extensions/variaions [7–10], and Correa et al. compared the results from Infomax, FastICA, JADE (Joint Approximate Diagonalization of Eigenmatrices), and EVD (Eigenvalue Decomposition)and found inconsistency between the results of EVD [14].

Some issues regarding the reliability of different ICA algorithms still remain. (1) The exact algorithmic reliability of different ICA algorithms is still unknown. Yang et al. assessed the algorithmic reliability of FastICA by using SCC [7]. Correa et al. applied ICASSO to evaluate the consistency of results from multiple runs of FastICA by using Infomax as a benchmark; however, they did not measure the reliability of Infomax [14]. No published reports examined the difference of different ICA algorithms with quantified index such as SCC (used by RAICAR) or Iq (provided by ICASSO). (2) ICA algorithms including AMUSE (Algorithm for Multiple Unkown Signal Extraction), JADE, ERICA (Extended Robust Independent Components Analysis), RADICAL (Robust, Accurate, Direct ICA aLgorithm), and SIMBEC produce identical results after each run (deterministic algorithms), whereas Infomax, FastICA, EVD, and COMBI (Combination of WASOBI and EFICA; WASOBI: Weights-Adjusted Second Order Blind Identification, EFICA: Efficient variant of algorithm FastICA) generate different results after repeated analysis (non-deterministic algorithms). How many times a non-deterministic algorithm should be run to acquire a reliable result remains unknown. Yang et al. [7] found that more than 20 times was not necessary, but only FastICA was tested.

ICASSO presents clusters after integrating the results of multiple runs. The estimates of an IC will be incorporated into a cluster if these estimates are consistent, and the number of estimates in a cluster is equivalent to the number of runs if the corresponding ICs are estimated consistently. Therefore, the number of estimates in each cluster can also be used to assess the reliability of an ICA algorithm.

This study aimed to solve three problems. (1) The Iq and the number of estimates in each cluster provided by ICASSO were used together to quantify the reliability of four non-deterministic algorithms (i.e., Infomax, FastICA, EVD, and COMBI). (2) Each of the four non-deterministic algorithms was repeated at different times with ICASSO (from 10 times to 100

times for each algorithm), and the SCC was used to determine how many times were needed to acquire ICA results with good reliability. (3) The SCC was used to quantify the consistency among the results of nine ICA algorithms (the four non-deterministic algorithms and the five deterministic algorithms, including AMUSE, JADE, ERICA, RADICAL, and SIMBEC) and determine the algorithm with the best reliability.

We explored real fMRI data involving sensory stimulation and motor task execution in this study.

## Materials and methods

### Subjects, stimulation/task paradigms, and image acquisition

The sensory stimulation data were acquired by using a 1.5T MRI scanner from a female patient (aged 18 years) with traumatic brain injury. The fMRI data were acquired with a gradient echo type echo planar imaging (GRE–EPI) sequence (Repetition Time = 3000 ms, Echo Time = 40 ms, flip angle = 90˚, field of view = 240 mm × 240 mm, matrix size = 64 × 64, slice thickness = 5 mm, no gap, 23 contiguous axial slices). In one fMRI session, a block design protocol was applied with successive blocks alternating between rest and transcutaneous electrical nerve stimulation, starting with rest. There was a 24 s dummy period before MR data collection. Each block duration of rest or electrical stimulation was 30 s. The rest–stimulation cycle was repeated six times. Thus, the total duration of the fMRI session was 6 min 24 s. The electrical stimulation was performed at a rate of 2 Hz with surface electrodes. The cathode was placed on the space between the spinous processes of L2 and L3, whereas the anode was placed on the space between the spinous processes of L4 and L5. The stimulation was performed with a WQ-10D stimulator (Beijing Electronics Instrument Co., Ltd.). The intensity was set to a score of 2 rated on a 0–10 verbal numerical rating scale. T1-weighted images were also acquired to provide an anatomical reference.

The motor task data were acquired by using the same scanner from a healthy male subject (aged 30 years). The fMRI data were acquired with a GRE–EPI sequence (Repetition Time = 3000 ms, Echo Time = 40 ms, flip angle = 90˚, field of view = 240 mm × 240 mm, matrix size = 64 × 64, slice thickness = 5 mm, no gap, 28 contiguous axial slices). T1-weighted images were also acquired. Two types of motor tasks were applied (i.e., imagined movements and motor execution task) in one fMRI session. The motor execution task consisted of 90˚ isometric dorsiflexion of the right foot. During the imagined movement task, the subject was required to imagine performing the same type of foot movement instead of actually executing it. The total duration of the fMRI session was 6 min. The experiment consisted of eight 21 s periods of baseline alternating with eight 21 s periods of motor tasks. The imagined movements were followed by the motor execution task in sequence. The beginning and ending of each task block were signaled with verbal commands "Ready, imaging, right, start" and "stop" or "Ready, move, right, start" and "stop." The duration of each command was 3 s. An audio cue repeated every 3 s was broadcasted during each task block to prompt the subject to persistently focus on the task. The two fMRI data sets contained three types of status, i.e., sensory stimulation, imagined movements and motor execution task. The fMRI data sets can be found in the S1 Data.

The experiment was conducted with the approval from the institutional review board of the National Research Center for Rehabilitation Technical Aids, and written informed consents were obtained from the subjects. The procedure of the experiment was in accordance with the principles of the Declaration of Helsinki.

## Performing ICA

The Statistical Parametric Mapping (SPM) software (https://www.fil.ion.ucl.ac.uk/spm) was used to perform motion correction, coregistration, and spatial smoothing with a Gaussian kernel of 10 mm full-width at half-maximum.

ICA was performed using the Group ICA of fMRI Toolbox (GIFT) software (http://mialab.mrn.org/software/gift/index.html). The number of ICs was estimated by using the minimum description length (MDL) criteria [15]. Except for the algorithm and the number of ICs, default settings/parameters defined by the GIFT software were used during analysis.

In group analysis containing a number of subjects, subjects in a group can be concatenated as a single ensemble by the GIFT software. In this study, there was a single ensemble for each subject since each data set contained only one subject. During data reduction steps, for one subject and one session, the data reduction actually would be disabled since the number of principal components extracted from the data is the same as the number of independent components.

## Assessing the reliability of four non-deterministic ICA algorithms with ICASSO

The ICASSO toolbox embedded in the GIFT software was employed to run 10 times with different repetition number $k$ ($k$ = 10, 20, 30, 40, 50, 60, 70, 80, 90, and 100) for each of the four non-deterministic algorithms.

The RandInit mode (algorithm starts with Randomizing different Initial values) was used. This mode in ICASSO uses the original data whereas the data will be resampled if the bootstrapping method is used. Additionally, the RandInit mode generates correlation coefficients with straightforward calculations whereas some extra normalization is necessary for bootstrapping [8]. When we run ICASSO 10 times by using the RandInit mode, the algorithm (e.g., Infomax) would run 10 times; each time the algorithm started with randomizing different initial conditions.

For a given algorithm, each of the 10 results of ICASSO contained $N$ clusters; the number of $N$ was determined by the MDL criteria. For an ICASSO result with a repetition number $k$, clusters presented by the ICASSO toolbox included estimated ICs from each run if these ICs presented high mutual similarities. The centrotype of a cluster holds the maximum sum of similarities to other points in the cluster and was used as the representation of this cluster by ICASSO. In ICASSO, the similarity between one pair of ICs ($i$ and $j$) is quantified by the absolute value of their mutual correlation coefficients $\sigma_{ij}$. The clustering process is performed by using the distance between the two ICs. The distance is determined by transforming the similarity matrix into a dissimilarity (distance) matrix: $d_{ij} = 1 - \sigma_{ij}$ [8].

The reliability of the four non-deterministic algorithms was assessed with two approaches. The Iq was used to be the main index of reliability. First, for each algorithm, we compared the Iq values among results with different repetition number $k$ to determine whether there is any difference when the repetition number $k$ increases and to find the best $k$ value with which a given algorithm could provide the most reliable result. Second, the number of ICs in each cluster will be equal to the repetition number $k$ if an ICA algorithm is reliable, whereas an unreliable algorithm will generate clusters containing less number of ICs than the repetition number $k$. Thus, we compared how many clusters contained different numbers of ICs from the repetition number $k$ for each of the four non-deterministic algorithms.

Afterward, we compared the Iq values between the most reliable results of the four non-deterministic algorithms to determine which ICA algorithm exhibited the highest reliability (i.e., the highest Iq value).

## Comparing the reliability among four non-deterministic ICA algorithms with SCC

The SCCs between each pair of ICs from each pair of ICASSO results using the same algorithm were calculated by using MATLAB function *corrcoef*. The SCC was the Pearson correlation coefficient value between spatial maps of a pair of ICs. Afterward, the SCCs were used in two steps.

Step 1 was a process to select the maximum value. For a non-deterministic algorithm, suppose that the total number of ICs in each ICASSO result (when $k$ = 10, 20, 30, 40, 50, 60, 70, 80, 90, and 100, the corresponding result was A, B, C, ..., J, respectively) is $i$. Then, the $j$-th IC map ($j$ = 1, 2, ..., $i$) in ICASSO result A would have $i$ SCC values with each IC map in ICASSO result B for this algorithm. The highest one among these $i$ SCC values indicates the best spatial consistency/similarity. Thus, an IC with the highest SCC value in ICASSO result B can be considered to match the $j$-th IC in ICASSO result A. In this way, each pair of matched ICs in each pair of ICASSO results was determined. As a result, we acquired a list of SCCs for each pair of groups.

In Step 2, for each non-deterministic algorithm, the SCC values between the most reliable result (determined in the former section) and the other nine results were compared by using the Kruskal–Wallis test to measure the impact of repetition time $k$ on the consistency of spatial maps.

## Comparing the reliability among nine ICA algorithms with SCC

Finally, we compared the results from the nine algorithms with the SCC value as an index to investigate the spatial difference of results acquired with different algorithms. For the non-deterministic ICA algorithms Infomax, FastICA, EVD, and COMBI, the most reliable results were used to represent the most approximation of the source.

## Results

Using the MDL criteria, the estimated numbers of ICs were 12 for the sensory stimulation data and 8 for the motor task data.

## The most reliable result for the four non-deterministic algorithms when using the Iq value as an index

**Sensory stimulation data.** The median Iq values of different results for each of the four non-deterministic algorithms are shown in **Fig 1**. The fluctuation of Infomax results appeared to be much less than those of the other algorithms. For each of the four algorithms, the ICASSO repetition number $k$ did not influence the Iq values (**Table 1**). For all of the four algorithms, the repetition number $k$ with the highest median is displayed in **Table 2**. The ICASSO results with such number $k$ were the most reliable ones for each non-deterministic algorithm.

**Motor task data.** The median Iq values of different results for each of the four non-deterministic algorithms are shown in **Fig 2**. The fluctuation of results of Infomax also appeared to be much less than those of the other three algorithms. For each of the four algorithms, the ICASSO repetition number $k$ did not influence the Iq values (**Table 3**). For all of the four algorithms, the repetition number $k$ with the highest median is displayed in **Table 4**.

Taken together, Infomax presented better reliability than the other three non-deterministic algorithms.

## Most reliable non-deterministic algorithm when using Iq values as the index

For each non-deterministic algorithm, the ICASSO result with the highest median Iq values was used as the most reliable one (i.e., those shown in **Table 2** for the sensory data and **Table 4**

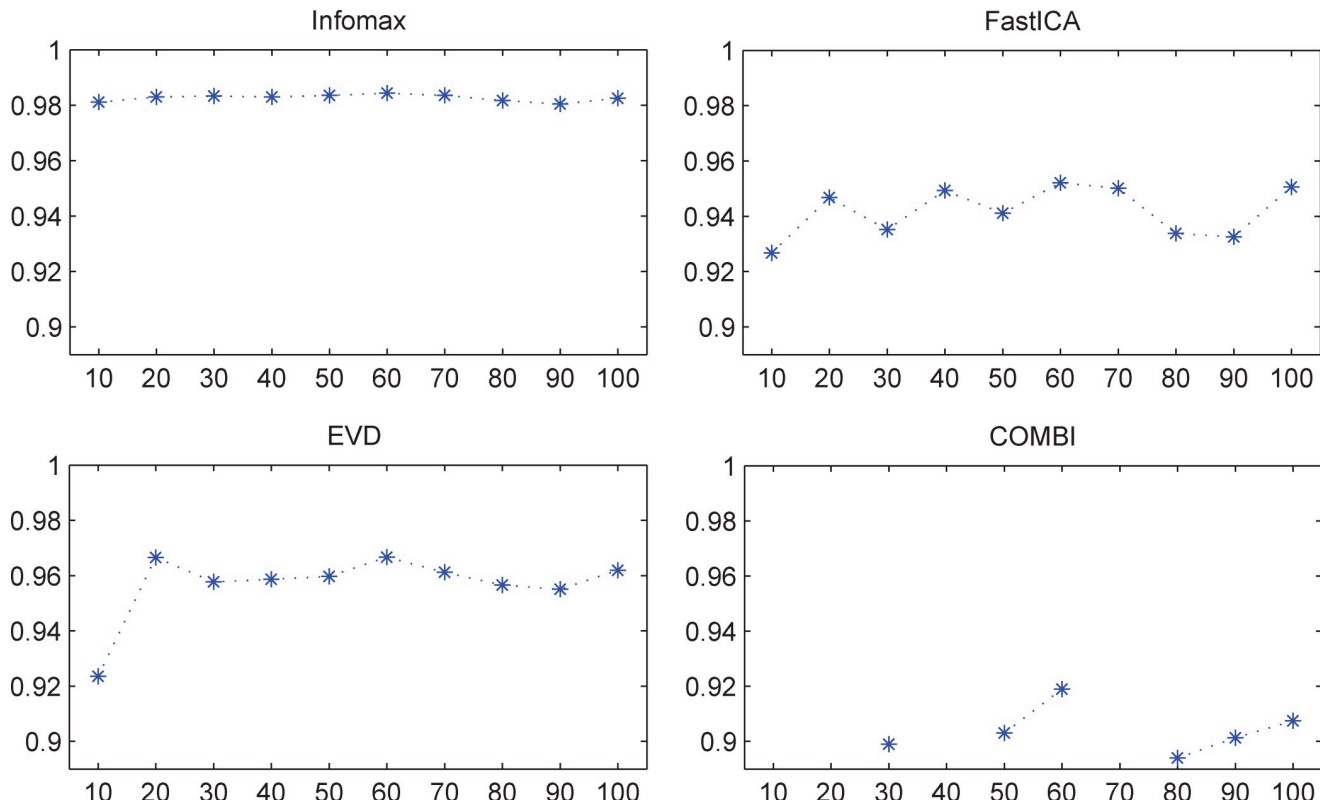

**Fig 1. Median Iq values of different results for non-deterministic ICA (sensory data).** The unit of the x-axis is the ICASSO repetition number $k$, and the unit of the y-axis is the Iq value. The results of COMBI presented clusters containing only one estimate when $k$ = 10, 20, 40, and 70; under such conditions, the corresponding Iq value could not be calculated, which suggested very poor reliability.

for the motor data). Here, these results were used to represent each algorithm and were then compared. The results of the Kruskal–Wallis test are shown in **Table 5**.

For both sensory and motor data, Infomax generated higher median Iq values than other algorithms although the difference was not statistically significant ($p$ = 0.1709, Table 5). For the motor data, the median Iq value of the most reliable results from EVD ($k$ = 80) was significantly lower than those of the other three non-deterministic algorithms. The result of COMBI was significantly lower than those of Infomax and FastICA when comparing the results from Infomax, FastICA, and COMBI (Chi-sq 15.7650, $p$ = 0.0003773, the medians shown in Table 4). The results of Infomax and FastICA were not statistically different (Chi-sq 0.8934, $p$ = 0.3446).

Thus, Infomax presented better reliability than the other three non-deterministic algorithms.

**Table 1. Repetition number $k$ did not influence Iq values for non-deterministic ICA (sensory data).**

|         | Infomax | FastICA | EVD   | COMBI  |
|---------|---------|---------|-------|--------|
| Chi-sq  | 1.04    | 1.11    | 11.67 | 0.8406 |
| P       | 0.9993  | 0.9992  | 0.2323| 0.9744 |

For each non-deterministic algorithm, the Iq values of different repetition number $k$ ($k$ = 10, 20, 30, 40, 50, 60, 70, 80, 90, and 100) did not present statistically significant difference (Kruskal–Wallis test). The result of the Kruskal–Wallis test of COMBI was based on six groups of Iq values because four results (when $k$ = 10, 20, 40, and 70) of COMBI had clusters without the Iq value.

**Table 2. ICASSO repetition number *k* with the highest median Iq of each algorithm (sensory data).**

|        | Infomax | FastICA | EVD    | COMBI  |
|--------|---------|---------|--------|--------|
| k      | 60      | 60      | 60     | 60     |
| median | 0.9843  | 0.9522  | 0.9667 | 0.9190 |

The result of COMBI was based on six groups of Iq values because four results (when *k* = 10, 20, 40, and 70) had clusters without the Iq value.

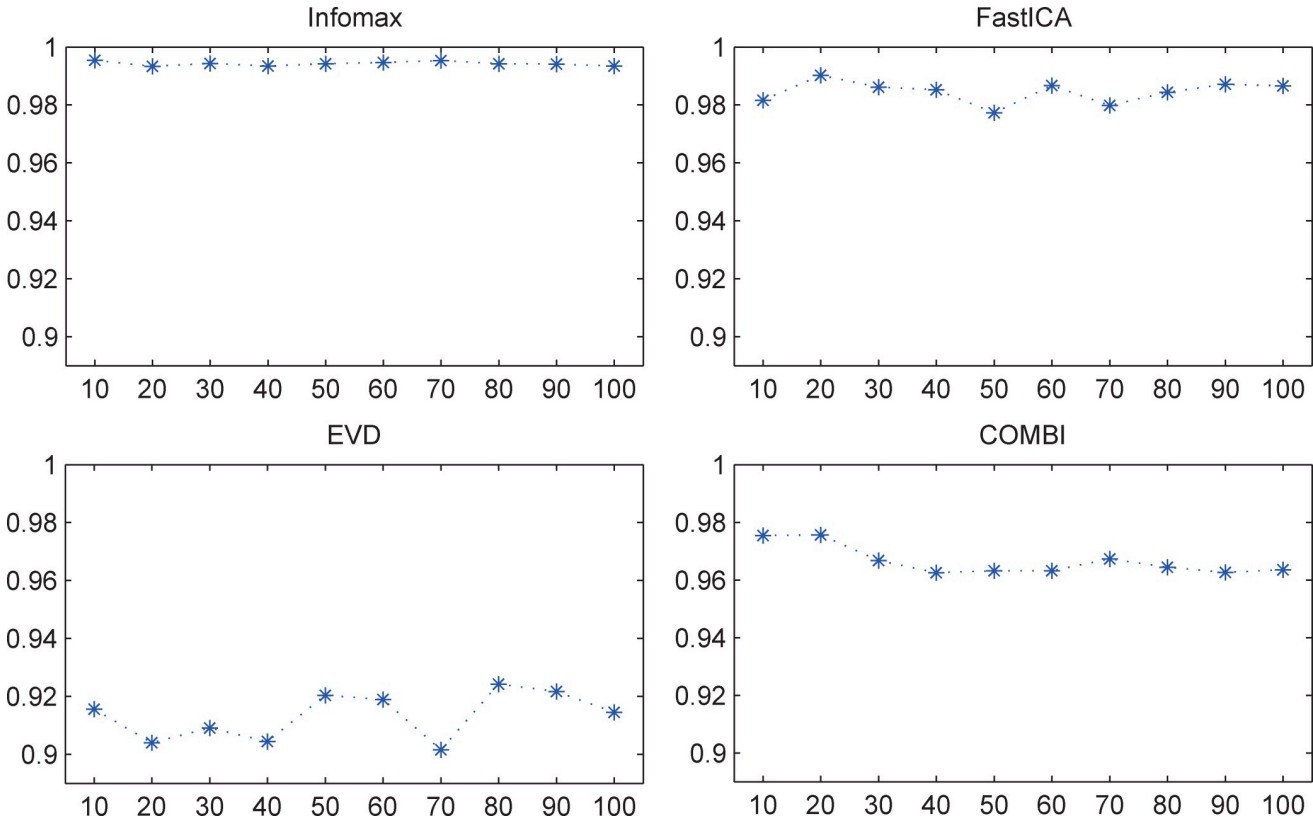

**Fig 2. Median Iq values of different results for non-deterministic ICA (motor data).** The unit of the x-axis is the ICASSO repetition number k, and the unit of the y-axis is the Iq value.

**Table 3. Repetition number *k* did not influence Iq values for non-deterministic ICA (motor data).**

|        | Infomax | FastICA | EVD    | COMBI  |
|--------|---------|---------|--------|--------|
| Chi-sq | 15.37   | 8.83    | 1.99   | 9.14   |
| P      | 0.0814  | 0.4535  | 0.9916 | 0.4241 |

For each non-deterministic algorithm, the Iq values of different repetition number *k* (*k* = 10, 20, 30, 40, 50, 60, 70, 80, 90, and 100) did not present statistically significant difference (Kruskal–Wallis test).

**Table 4. ICASSO repetition number *k* with the highest median Iq of each algorithm (motor data).**

|        | Infomax | FastICA | EVD    | COMBI  |
|--------|---------|---------|--------|--------|
| k      | 10      | 20      | 80     | 20     |
| median | 0.9954  | 0.9902  | 0.9242 | 0.9756 |

Table 5. Differences among the most reliable results of each non-deterministic algorithm.

|         | Sensory data | Motor data |
|---------|--------------|------------|
| Chi-sq  | 5.0128       | 22.0256    |
| P       | 0.1709       | 0.00006    |

## Assessing the reliability using the number of clusters containing different numbers of ICs from ICASSO repetition number $k$ as an index

**Sensory stimulation data.** For Infomax, all clusters of each ICASSO result contained the same number as the repetition number $k$; for FastICA, each ICASSO result contained at least two clusters presenting different numbers of ICs from ICASSO repetition number $k$ when $k > 30$, and the total number of such clusters was 21; for EVD, the ICASSO results contained two clusters presenting different numbers of ICs from the repetition number $k$ when $k = 80$; for COMBI, each ICASSO result contained at least two clusters presenting different numbers of ICs from the repetition number $k$, and the total number of such clusters was 48.

**Motor task data.** For Infomax, all clusters of each ICASSO result contained the same number as the repetition number $k$; for FastICA, when $k = 50, 70$, and 80, each ICASSO result contained two clusters presenting different numbers of ICs from the repetition number $k$, and the total number of such clusters was 6; for EVD, when $k = 90$, the ICASSO results contained two clusters presenting different IC numbers from the repetition number $k$; for COMBI, when $k > 50$, each ICASSO result contained at least two clusters presenting different numbers of ICs from the repetition number $k$, and the total number of such clusters was 13.

Taken together, Infomax exhibited the best reliability when using the number of clusters containing IC numbers different from the ICASSO repetition number $k$ as an index.

## Difference in SCC values between the most reliable results and the other results for each non-deterministic algorithm

For each algorithm, no statistical difference in SCC values was found between the most reliable ICASSO results and each of the other nine results for either sensory data (**Table 6**) or motor data (**Table 7**).

The range of SCC values for each non-deterministic algorithm is presented in **Tables 8** and **9**. Note for the sensory data, FastICA and COMBI had very low SCC values, indicating poor spatial reliability.

For the sensory data, if the SCC values $\leq 0.88$ (found in the results of COMBI), there would be an IC (in the corresponding ICASSO result) that matched two IC maps of the most reliable result (i.e., presenting similar SCC values), and there was another IC (in the corresponding ICASSO result) that did not match any IC in the most reliable result. In other words, when

Table 6. Differences in SCC values between the most reliable ICASSO results and the other nine results (sensory data).

|         | Infomax | FastICA | EVD    | COMBI  |
|---------|---------|---------|--------|--------|
| Chi-sq  | 2.3734  | 0.7240  | 6.1818 | 5.0999 |
| P       | 0.9674  | 0.9995  | 0.6269 | 0.7468 |

For each non-deterministic algorithm, ICASSO was run $k$ times ($k = 10, 20, 30, 40, 50, 60, 70, 80, 90, 100$) to acquire 10 results. The most reliable ICASSO results (shown in Table 2 for the sensory data) and the other results were compared by using the Kruskal–Wallis test.

**Table 7. Differences in SCC values between the most reliable ICASSO results and the other nine results (motor data).**

|  | Infomax | FastICA | EVD | COMBI |
|---|---|---|---|---|
| Chi-sq | 3.2546 | 4.0372 | 7.0266 | 2.9837 |
| P | 0.9174 | 0.8537 | 0.5338 | 0.9354 |

For each non-deterministic algorithm, ICASSO was run *k* times (*k* = 10, 20, 30, 40, 50, 60, 70, 80, 90, 100) to acquire 10 results. The most reliable ICASSO results (shown in Table 4 for the motor data) and the other results were compared by using the Kruskal–Wallis test.

compared with the most reliable result, the other nine results with SCC ≤ 0.88 indicated unreliable performance. Thus, we set 0.88 as a threshold to numbers of such SCC values in **Tables 8** and **9** (using 0.9 referring to [7]).

Taken together, Infomax exhibited the best reliability.

## Comparing SCC values among the nine ICA algorithms

In previous sections, Infomax always presented better reliability than other non-deterministic algorithms. Infomax is a widely used ICA algorithm. It has been proven to be quite reliable for fMRI data analysis [6, 14, 16] and is a well-performing algorithm for other data types [17, 18]. Thus, we used ICASSO results of Infomax as a reference to compare the performance of other ICA algorithms against Infomax. For either sensory data or motor data, SCC values were computed between Infomax (the most reliable ICASSO result) and the results of the other eight ICA algorithms. For FastICA, EVD, and COMBI, the most reliable ICASSO results were used.

For both sensory data and motor data, only the results of FastICA exhibited good spatial consistency with the results of Infomax when considering median and minimum SCC values (**Tables 10** and **11**). Other algorithms had very low minimum SCC values in the results of either sensory or motor data, or both of them.

If there was an SCC value less than 0.669816, there would be at least one "unmatched" IC in the result of such algorithm, which means that this IC could not exclusively match any IC map in the results of Infomax and thus suggests a poor spatial consistency between the results of the algorithm and Infomax.

## Discussion

Whichever index was used (Iq values, cluster numbers, and SCC values), Infomax always presented better reliability than other non-deterministic algorithms (FastICA, EVD, and COMBI). Infomax also exhibited better consistency than the five deterministic algorithms (AMUSE, JADE, ERICA, RADICAL, and SIMBEC) when using SCC as the index. Compared with Infomax, FastICA presented weaker consistency with some unreliable separated ICs. Other algorithms exhibited poor consistency under some conditions.

**Table 8. Range of SCC values between the most reliable ICASSO results and the other nine results (sensory data).**

|  | Infomax | FastICA | EVD | COMBI |
|---|---|---|---|---|
| Max | 1 | 0.999999 | 0.999357 | 1.000000 |
| Min | 0.969351 | 0.005471 | 0.994896 | 0.441637 |
| Number of clusters with SCC<0.9 | 0 | 5 | 0 | 7 |

**Table 9. Range of SCC values between the most reliable ICASSO results and the other nine results (motor data).**

| | Infomax | FastICA | EVD | COMBI |
|---|---|---|---|---|
| Max | 1 | 1 | 1 | 1 |
| Min | 0.998238 | 0.997670 | 0.998731 | 0.998982 |
| Number of clusters with SCC<0.9 | 0 | 0 | 0 | 0 |

When running Infomax with ICASSO, repetition times > 10 did not lead to significant differences between results, indicating that 10 times can be used to acquire reliable ICs when using this algorithm. In additional to the high reliability of Infomax, this finding can also be attributed to the algorithm of ICASSO that generates centrotype of each cluster as the representative. The centrotype is calculated as the maximum sum of similarities to other points in a cluster [8], and running Infomax with ICASSO can therefore generate consistent results after a few repetition times.

When using ICA to explore cerebral networks, especially for conditions with low signal-to-noise ratio such as resting state [19], consistent ICA results generated by using a reliable algorithm such as Infomax can lead to higher study efficiency. Otherwise, variations of ICA results from different runs enhance the difficulty in explaining acquired ICs.

Our results demonstrate that Infomax running 10 times with ICASSO can generate consistent ICs from fMRI data sets. This finding provides an easily accessible approach to generate reliable ICs. These ICs can be used to develop advanced analyses such as personalized brain networks [20].

The results also demonstrate that the algorithms other than Infomax produce more or less unreliable separated ICs. Therefore, caution needs to be taken in explaining ICA results when the consistency of ICs cannot be assured.

The SCC was the correlation coefficient value between two IC spatial maps, that is, two spatial matrices. When comparing two groups of ICs A and B, the correlation coefficient value between each IC in group A and each IC in group B was calculated. The best-matched ICs must be a pair presenting the highest correlation value. Such a pair was unique since the corresponding correlation value was the highest one, and we did not find an IC in one group presenting two equally highest correlation values with two different ICs in the other group. We used the correlation coefficient value to compare different conditions, and the correlation values can be transformed into normal variables by using the Fisher-Z transform. Suppose that the population of correlation coefficient values is $\rho$, and the sample of correlation coefficients is $r$ (the sample is a part of the population). When the Fisher-Z transformation is applied to the sample $r$, the sampling distribution of the transformed variable is approximately normal. Without the Fisher transformation, the variance of $r$ (between two variables $X$ and $Y$) grows smaller as $|\rho|$ gets closer to 1 and thus is not normally distributed. In this study, however, for each comparison between two groups of SCCs, the normal distribution of the transformed

**Table 10. SCC values between the most reliable Infomax results and the other nine results (sensory data).**

| | AMUSE | ERICA | JADE | RADICAL | SIMBEC | FastICA | EVD | COMBI |
|---|---|---|---|---|---|---|---|---|
| Median | 0.708041 | 0.656005 | 0.95573 | 0.910922 | 0.685752 | 0.9972918 | 0.51767 | 0.991084 |
| Max | 0.916648 | 0.992157 | 0.998284 | 0.996636 | 0.992512 | 0.9999235 | 0.819438 | 0.999003 |
| Min | 0.519998 | 0.279998 | 0.096291 | 0.033701 | 0.186971 | 0.9786125 | 0.244201 | 0.018344 |

The results of FastICA (the most reliable results) presented higher SCC values than other algorithms.

**Table 11. SCC values between the most reliable Infomax results and the other nine results (motor data).**

|  | AMUSE | ERICA | JADE | RADICAL | SIMBEC | FastICA | EVD | COMBI |
|---|---|---|---|---|---|---|---|---|
| Median | 0.801476 | 0.75061 | 0.94187 | 0.98748 | 0.739344 | 0.998879 | 0.623538 | 0.99123 |
| Max | 0.977052 | 0.939287 | 0.999795 | 0.998379 | 0.992799 | 0.9998937 | 0.97587 | 0.999588 |
| Min | 0.652854 | 0.476254 | 0.669816 | 0.723642 | 0.491713 | 0.963321 | 0.243115 | 0.88953 |

SCC values in each group cannot be assured. The reason is that the SCC values ($r$ values) come from different populations. For example, the SCC value between $j$-th IC in group A and $k$-th IC in group B can be Fisher-Z transformed, which leads to a normal distribution of the transformed SCC value. Such a normal distribution means that the transformed $r$ values meet normal distribution for the two populations where two samples ($j$-th IC in group A and $k$-th IC in group B) are drawn. But now we have other SCC values between other pairs of ICs (e.g., $m$-th IC in group A and $n$-th IC in group B). These are samples from other populations. Therefore, the transformed $r$ values may not be normally distributed when put into a group. As the solution, we used the Kruskal–Wallis test in Tables 6 and 7 since the Kruskal–Wallis test is a nonparametric test and does not assume the normality of the measurement variable.

This study used fMRI data from two subjects. These fMRI data sets contained three types of status, i.e., sensory stimulation, imagined movements, and motor execution task. Single-subject ICA analysis is the foundation of group-level analysis. However, results from two single-subject analyses may not well represent the performance of an ICA algorithm for all types of data. Whether the reliability of Infomax is superior to other algorithms needs to be further proved with other types of data.

## Supporting information

**S1 Data.**
(RAR)

## Acknowledgments

We appreciate the suggestions of reviewers and the clarification of the origin of the word ICASSO by Prof. Aapo Hyvärinen (that is, ICASSO is not an acronym but evolved from the acronym ICA and the name of the artist Picasso).

## Author Contributions

**Conceptualization:** Pengxu Wei, Yubo Fan.

**Data curation:** Pengxu Wei, Ruixue Bao.

**Formal analysis:** Pengxu Wei.

**Funding acquisition:** Pengxu Wei.

**Investigation:** Pengxu Wei, Ruixue Bao.

**Methodology:** Pengxu Wei.

**Validation:** Pengxu Wei, Yubo Fan.

**Writing – original draft:** Pengxu Wei, Yubo Fan.

**Writing – review & editing:** Pengxu Wei, Yubo Fan.

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
