## [Decision Letter · Decision Letter 0]

25 Jan 2022

PONE-D-21-19322Comparing the reliability of different ICA algorithms for fMRI analysisPLOS ONE

Dear Dr. Wei, Thank you for submitting your manuscript to PLOS ONE. After careful consideration, we feel that it has merit but does not fully meet PLOS ONE’s publication criteria as it currently stands. Therefore, we invite you to submit a revised version of the manuscript that addresses the points raised during the review process.

Sorry for the long review process so far, and we had only one reviewer's feedback, the experts comments are comprehensive and in-depth. I feel it is fair to move forward with you starting the revision process and I personally went over the MS couple time and concur with the reviewer.

We look forward to receiving your revised manuscript.

Kind regards,

Kewei Chen, Ph.D

Academic Editor

PLOS ONE

Journal Requirements:

"This research was funded by the National Key R&D Program of China (Grant Nos. 2018YFC2001400 and 2018YFC2001700), the National Natural Science Foundation of China (Grant No. 81972160), and the Beijing Natural Science Foundation (Grant No. 17L20019)."

"This research was funded by the National Key R&D Program of China (Grant Nos. 2018YFC2001400 and 2018YFC2001700 by WP and ZL), the National Natural Science Foundation of China (Grant No. 81972160 by WP), and the Beijing Natural Science Foundation (Grant No. 17L20019 by WP).

6. Please include a copy of Table S8, S9, S2 and S4 which you refer to in your text on pages 6, 11 and 12.

Reviewers' comments:

Reviewer's Responses to Questions

**Comments to the Author**

1. Is the manuscript technically sound, and do the data support the conclusions?

Reviewer #1: Yes

2. Has the statistical analysis been performed appropriately and rigorously? 

Reviewer #1: No

3. Have the authors made all data underlying the findings in their manuscript fully available?

Reviewer #1: No

4. Is the manuscript presented in an intelligible fashion and written in standard English?

Reviewer #1: Yes

5. Review Comments to the Author

Reviewer #1: This is an interesting article. There is real value. However, I have several concerns regarding details missing, and technical questions that are unanswered in the article in its present form. The article is generally well written, but all acronyms are not fully spelled out on first use, and software packages and links are not always provided in the methods. The authors would have made both review and their revision tasks easier if they had added line numbers in the draft.

I will detail the various data and methodology concerns in detail below.

1. ACRONYMS must be spelled out on first use and links to software and citations provided. The omissions are numerous. A few (not complete list) MDL, AMUSE, JADE, ERICA and surprisingly, even ICASSO which is a pivotal method here.

2. Only two human subject data sets are used in this paper. This might be acceptable if we were convinced that the data was sufficiently rich, but this information is missing. The reader needs to know fully data set sizes: numbers of imaging sessions, full number of tasks, full number of stimuli in tasks etc. The richness of data is a factor in ICA outcomes so these aspects MUST be reported. Similarly, the method of data setup for ICA needs better description. Was data analyzed in each subject ultimately as a single concatenated ensemble (and if so, of what matrix dimensions), or as a number of subset ensembles (and if so, of what dimensions, and how many such ensembles). Because only two individuals data were used, there remains the unfortunate possibility that there are structural individual-based idiosyncracies in the data here that favor Infomax. I do not believe this likely, but it cannot be fully discounted given the data. Other research applying ICA or testing dimensionality separation methods including ICA have used larger numbers of individuals, or/and artificial data generation under constraints to explore comparisons, or reliability. Examples in motor field are Tresch, D'Avella, Cheung, papers in J Neurophysiology for synergy separation analyses, and Yang, Logan and Giszter in PNAS exploring SCC-like measures on motor synergy outcomes across individual animals. An aspect not explored by the authors of this paper (and given lack of detail provided, this may or may not be possible in their data), that might be used to enrich the analysis they perform is bootstrapping or jack-knifing the data sets used, in addition to subsequent multiple iterations of ICA methods on each subjects subset data. At the very least, there needs to be discussion of the data limits in this paper resulting from only 2 subjects and only 1 subject per type of experiment, and the caveats resulting, i.e. possibility of individual idiosyncracy of data favoring specific ICA methods.

3. The SCC method as described in section 2.4 is unclear. It is possible to do this in two ways - correlate the individual IC spatial components picking best correlations (where pairings of and IC with other sets may not be unique- same IC best in two or more correlations), or correlate the spatial matrix (e.g., in MATLAB using matperm, matcorr) and then IC correlations will be unique, based on the matrix permutation use in the unique IC matchings. The authors used the former, non-unique method I believe, but this might have biased results. If they did, with the possibility of non-unique matching of ICs, then I think the reader needs to know if there were non-unique best IC spatial correlations in each method, and if so how many. This might be an additional metric on ICA algorithm quality.

4. Finally, in relation to the SCC statistics used: Correlation, especially here, is non-gaussian if used directly. This may present difficulties in interpreting the statistics in tables 6 and 7 that could be avoided. Using the Fisher-Z transform (see Yang, Logan, Giszter, PNAS 2019 for example with Infomax ICA data stats) the correlations are transformed to normal variables, improving the interpretation of parametric methods.

5. Equations would be helpful throughout the methods if very clearly written. The goal is reproducibility of methods.

6. The data sharing statement is insufficient, for this study an anonymized repository should be chosen - there are many.

This paper is a solid contribution, but marred by the omissions in detail, and possibly improved by attention to the technical points noted.

6. PLOS authors have the option to publish the peer review history of their article (what does this mean?). If published, this will include your full peer review and any attached files.

Reviewer #1: No

---

## [Author Response · Author response to Decision Letter 0]

3 Feb 2022

"This research was funded by the National Key R&D Program of China (Grant Nos. 2018YFC2001400 and 2018YFC2001700), the National Natural Science Foundation of China (Grant No. 81972160), and the Beijing Natural Science Foundation (Grant No. 17L20019)."

"This research was funded by the National Key R&D Program of China (Grant Nos. 2018YFC2001400 and 2018YFC2001700 by WP and ZL), the National Natural Science Foundation of China (Grant No. 81972160 by WP), and the Beijing Natural Science Foundation (Grant No. 17L20019 by WP).

6. Please include a copy of Table S8, S9, S2 and S4 which you refer to in your text on pages 6, 11 and 12.

We prepared the new version on the basis of “Journal Requirements” mentioned in the Decision letter:

1. We referred to the The PLOS ONE style templates.

2. The correct grant numbers for the Funding are those in the Acknowledgments Section of the first version. That is, "This research was funded by the National Key R&D Program of China (Grant Nos. 2018YFC2001400 and 2018YFC2001700), the National Natural Science Foundation of China (Grant No. 81972160), and the Beijing Natural Science Foundation (Grant No. 17L20019)."

3. We have removed any funding-related text from the manuscript. Please move the following contents to the Funding Statement: "This research was funded by the National Key R&D Program of China (Grant Nos. 2018YFC2001400 and 2018YFC2001700), the National Natural Science Foundation of China (Grant No. 81972160), and the Beijing Natural Science Foundation (Grant No. 17L20019)."

4. The minimal data set has been included in the Supporting information.

5. The information on IRB has been added.

6. Table S8, S9, S2 and S4 should be Table 8, 9, 2 and 4. We have corrected the error in the revised version.

7. We have included captions for the Supporting Information files at the end of the manuscript, and updated in-text citations.

Reviewer #1: This is an interesting article. There is real value. However, I have several concerns regarding details missing, and technical questions that are unanswered in the article in its present form. The article is generally well written, but all acronyms are not fully spelled out on first use, and software packages and links are not always provided in the methods. The authors would have made both review and their revision tasks easier if they had added line numbers in the draft.

Response：We really appreciate the comments and suggestions. The revised manuscript, we hope, can solve the raised issues and is clearly expressed.

I will detail the various data and methodology concerns in detail below.

1. ACRONYMS must be spelled out on first use and links to software and citations provided. The omissions are numerous. A few (not complete list) MDL, AMUSE, JADE, ERICA and surprisingly, even ICASSO which is a pivotal method here.

Response: We did not find the origin of the acronym “ICASSO” when we prepare the manuscript. The developers provided a link http://research.ics.aalto.fi/ica/icasso/ where two publications (http://research.ics.aalto.fi/ica/icasso/publications.shtml) can be found. However, neither the content on the website nor the publications introduced the origin of ICASSO. We also failed to find the origin of the acronym in other publications. To find the answer, we asked the developer Prof. Aapo Hyvärinen and learn: it is not a real acronym. It is a kind of a joke on the acronym "ICA" and the name of the artist Picasso. We think this information is useful and have supplemented it in the Acknowledgments since no answer can be found publicly elsewhere.

The origins of other acronyms, links to software, and citations have been added.

2. Only two human subject data sets are used in this paper. This might be acceptable if we were convinced that the data was sufficiently rich, but this information is missing. The reader needs to know fully data set sizes: numbers of imaging sessions, full number of tasks, full number of stimuli in tasks etc. The richness of data is a factor in ICA outcomes so these aspects MUST be reported. Similarly, the method of data setup for ICA needs better description. Was data analyzed in each subject ultimately as a single concatenated ensemble (and if so, of what matrix dimensions), or as a number of subset ensembles (and if so, of what dimensions, and how many such ensembles). Because only two individuals data were used, there remains the unfortunate possibility that there are structural individual-based idiosyncracies in the data here that favor Infomax. I do not believe this likely, but it cannot be fully discounted given the data. Other research applying ICA or testing dimensionality separation methods including ICA have used larger numbers of individuals, or/and artificial data generation under constraints to explore comparisons, or reliability. Examples in motor field are Tresch, D'Avella, Cheung, papers in J Neurophysiology for synergy separation analyses, and Yang, Logan and Giszter in PNAS exploring SCC-like measures on motor synergy outcomes across individual animals. An aspect not explored by the authors of this paper (and given lack of detail provided, this may or may not be possible in their data), that might be used to enrich the analysis they perform is bootstrapping or jack-knifing the data sets used, in addition to subsequent multiple iterations of ICA methods on each subjects subset data. At the very least, there needs to be discussion of the data limits in this paper resulting from only 2 subjects and only 1 subject per type of experiment, and the caveats resulting, i.e., possibility of individual idiosyncracy of data favoring specific ICA methods.

Response: The fMRI data sets contained three types of status, sensory stimulation, imagined movement and motor execution task. The imagined movement is actually one type of cognitive task. We used these data to represent several different types. Based on the suggestion, missing information such as numbers of imaging sessions has been added.

More details of the method of data setup for ICA have been added. Except the algorithm and the number of ICs, default settings/parameters defined by the GIFT software were used during analysis. 

There was a single ensemble for each subject since each data set contained only one subject. During data reduction steps, for one subject one session, the data reduction actually would be disabled since the number of principal components extracted from the data is the same as the number of independent components, as introduced by the manual of the GIFT software, i.e., the matrix dimensions would not be changed. (In group analysis containing a number of subjects, subjects in a group can also be concatenated as a single ensemble by the GIFT software).

As you pointed out, some published studies such as Tresch, D'Avella, Cheung’s and Yang, Logan and Giszter’s have successfully performed ICA with Infomax. We suppose that many study groups have found some merits of Infomax empirically. The merit of this study is, on the aspect of reliability, to show the priority of Infomax and to uncover in which index Infomax is superior to other tested ICA algorithms in the single-subject level.

We used the RandInit mode (algorithm starts with Randomizing different Initial values) to run ICASSO. This information has been added in the manuscript. The RandInit mode was chosen because 1) the RandInit mode in ICASSO uses the original data whereas the data will be resampled in the bootstrapping method; 2) the RandInit mode generates correlation coefficients with straightforward calculations whereas some extra normalization is necessary for bootstrapping [J. Himberg, A. Hyvärinen and F. Esposito. NeuroImage 2004(3):1214-1222.]. If we run ICASSO 10 times, the algorithm (e.g., Infomax) will run 10 times; in each time, the algorithm starts with randomizing different initial conditions.

This study used fMRI data from two subjects. Discussions of this limitation have been added at the end of the manuscript. These fMRI data sets contained three types of status, i.e., sensory stimulation, imagined movements, and motor execution task. Single-subject ICA analysis is the foundation of group-level analysis. However, results from two single-subject analyses may not well represent the performance of an ICA algorithm for all types of data. Whether the reliability of Infomax is superior to other algorithms needs to be further proved with other types of data.

3. The SCC method as described in section 2.4 is unclear. It is possible to do this in two ways - correlate the individual IC spatial components picking best correlations (where pairings of and IC with other sets may not be unique- same IC best in two or more correlations), or correlate the spatial matrix (e.g., in MATLAB using matperm, matcorr) and then IC correlations will be unique, based on the matrix permutation use in the unique IC matchings. The authors used the former, non-unique method I believe, but this might have biased results. If they did, with the possibility of non-unique matching of ICs, then I think the reader needs to know if there were non-unique best IC spatial correlations in each method, and if so how many. This might be an additional metric on ICA algorithm quality.

Response: The SCC was the correlation coefficient value between two IC spatial maps, that is, two spatial matrices. When comparing two groups of ICs A and B, correlation coefficient value between each IC in group A and each IC in group B was calculated.

The best matched ICs must be a pair showing the highest correlation value. This best- matched pair is unique since the corresponding r value is the highest one; we did not find one IC in one group with two equal highest r values with two ICs in the other group. 

In our data, we only found lower thresholds: 1) (section 3.4 Difference in SCC values between the most reliable results and the other results for each non-deterministic algorithm) For the sensory data, if the SCC values ≤ 0.88 (found in the results of COMBI), there would be an IC (in the corresponding ICASSO result) that matched two IC maps of the most reliable result (i.e., presenting similar SCC values), and there was another IC (in the corresponding ICASSO result) that did not match any IC in the most reliable result. In other words, when compared with the most reliable result, the other nine results with a SCC ≤ 0.88 indicated unreliable performance; 2) (section 3.5 Comparing SCC values among the nine ICA algorithms). If there was an SCC value less than 0.669816, there would be at least one “unmatched” IC in the result of such algorithm, which means that this IC could not exclusively match any IC map in the results of Infomax and thus suggests a poor spatial consistency between the results of the algorithm and Infomax. This information has been presented in the manuscript.

4. Finally, in relation to the SCC statistics used: Correlation, especially here, is non-gaussian if used directly. This may present difficulties in interpreting the statistics in tables 6 and 7 that could be avoided. Using the Fisher-Z transform (see Yang, Logan, Giszter, PNAS 2019 for example with Infomax ICA data stats) the correlations are transformed to normal variables, improving the interpretation of parametric methods.

Response: We used correlation coefficient value to compare different conditions. The correlation values can be transformed to normal variables with the Fisher-Z transform.

The population correlation coefficient is ρ, and the sample correlation coefficient is r. The sample is a part of the population.

When Fisher-Z the transformation is applied to the sample correlation coefficient r, the sampling distribution of the transformed variable is approximately normal. Without the Fisher transformation, the variance of r (between two variables X and Y) grows smaller as |ρ| gets closer to 1 and thus is not normally distributed. (referring to publications listed on https://en.wikipedia.org/wiki/Fisher_transformation)

In our case, however, for each comparison between two groups of SCC, normal distribution of transformed SCC values in each group cannot be assured. The reason is that the SCC values (r values) come from different populations: For example, a SCC value between j-th IC in group A and k-th IC in group B can be Fisher-Z transformed, which leads to a normal-distributed transformed SCC value. Such normal distribution means the transformed r values meet normal distribution for the two populations where two samples (j-th IC in group A and k-th IC in group B) are drawn. But now we have other SCC values between other pairs of ICs (e.g., m-th IC in group A and n-th IC in group B). These are samples from other populations. Therefore, the transformed r values may not be normally distributed when putted into a group. For a large data set it may be possible (normally distributed), but we have only a small sample size here (12 ICs for the sensory data and 8 ICs for motor data). As the solution, we used the Kruskal–Wallis test in Table 6 and 7. The Kruskal–Wallis test is a nonparametric test that does not rely on normal distribution.

5. Equations would be helpful throughout the methods if very clearly written. The goal is reproducibility of methods.

Response: We supplemented an equation in section 2.3:

In ICASSO, the similarity between one pair of ICs (i and j) is quantified by the absolute value of their mutual correlation coefficients �i j. The clustering process is performed by using distance between the two ICs. The distance is determined by transforming the similarity matrix into a dissimilarity (distance) matrix: di j=1-�i j.

There are two commonly used methods to transform the similarity matrix into a distance matrix: di j=1-�i j or di j=1/�i j. Therefore, the supplemented information is helpful if someone want to verify the results since the two methods generate different values. There are some long and complex equations in the cited literature. We do not present these equations in that they need detailed introductions and may interfere with the topic of the manuscript.

6. The data sharing statement is insufficient, for this study an anonymized repository should be chosen - there are many.

Response: We have added the fMRI data to the Supporting information so that other groups can verify the results.

This paper is a solid contribution, but marred by the omissions in detail, and possibly improved by attention to the technical points noted.

Response: Thanks very much again for the comments and suggestions. We hope the new version responds well to the concerns.

---

## [Decision Letter · Decision Letter 1]

16 Mar 2022

PONE-D-21-19322R1Comparing the reliability of different ICA algorithms for fMRI analysisPLOS ONE

Dear Dr. Wei,

Thank you for submitting your manuscript to PLOS ONE. After careful consideration, we feel that it has merit but does not fully meet PLOS ONE’s publication criteria as it currently stands. Therefore, we invite you to submit a revised version of the manuscript that addresses the points raised during the review process.

We look forward to receiving your revised manuscript.

Kind regards,

Kewei Chen, Ph.D

Academic Editor

PLOS ONE

Journal Requirements:

Reviewers' comments:

Reviewer's Responses to Questions

**Comments to the Author**

1. If the authors have adequately addressed your comments raised in a previous round of review and you feel that this manuscript is now acceptable for publication, you may indicate that here to bypass the “Comments to the Author” section, enter your conflict of interest statement in the “Confidential to Editor” section, and submit your "Accept" recommendation.

Reviewer #1: All comments have been addressed

2. Is the manuscript technically sound, and do the data support the conclusions?

Reviewer #1: Yes

3. Has the statistical analysis been performed appropriately and rigorously? 

Reviewer #1: Yes

4. Have the authors made all data underlying the findings in their manuscript fully available?

Reviewer #1: Yes

5. Is the manuscript presented in an intelligible fashion and written in standard English?

Reviewer #1: No

6. Review Comments to the Author

Reviewer #1: This is a nice revision. Thank you for the work on the term ICASSO origin. This revision is coupled with a very responsive letter, but not all comments in the letter appear in the revised manuscript as I read it. Since readers may have the same concerns as reviewers it is crucial to include these in the manuscript, unless the full review history is also going to be published. In particular, the issue I could not find in the materials and methods:

That the component correlation method used (in section "Comparing the reliability among four ICA algorithms with SCC") did not in the present data produce any double use correlations must be stated, and that this was observed to be true, although this is not guaranteed by the method. I still wasn't clear if the authors were relying here on other code or directly checked this in their work, and this should be clearly stated.

Line 331 - probably, term 'well-performing' is better than 'well-performed' here

7. PLOS authors have the option to publish the peer review history of their article (what does this mean?). If published, this will include your full peer review and any attached files.

Reviewer #1: No

---

## [Author Response · Author response to Decision Letter 1]

28 Mar 2022

Thank you very much for the comments and suggestions. The explanations in the former letter have been added in the manuscript if the contents were not included in the former version.

In section "Comparing the reliability among four non-deterministic ICA algorithms with SCC” (and other sections where SCCs were calculated), the SCC was calculated by using MATLAB function corrcoef. We have added this information in the manuscript.

In this section, the SCC values are used in two steps. 

Step 1 is a simple picking-the maximum-value process, introduced in the second paragraph of this section. This is not a statistical comparison but just picks the highest SCC value. As a result, we get a list of SCC values for each pair of groups.

Step 2 is a statistical comparison between different lists of SCCs by using the Kruskal–Wallis test, introduced in the last paragraph of this section.

Thus, this is not a double use of correlations. Step 1 provides values for statistical analysis in Step 2.

In the new version, we have added this information and highlighted supplemented words. We hope the contents will be more clearly expressed.

The word 'well-performed' has been changed to 'well-performing' based on your suggestion.

---

## [Decision Letter · Decision Letter 2]

14 Jun 2022

Comparing the reliability of different ICA algorithms for fMRI analysis

PONE-D-21-19322R2

Dear Dr. Wei,

We’re pleased to inform you that your manuscript has been judged scientifically suitable for publication and will be formally accepted for publication once it meets all outstanding technical requirements.

Kind regards,

Pew-Thian Yap

Academic Editor

PLOS ONE

Additional Editor Comments (optional):

Reviewers' comments:

Reviewer's Responses to Questions

**Comments to the Author**

1. If the authors have adequately addressed your comments raised in a previous round of review and you feel that this manuscript is now acceptable for publication, you may indicate that here to bypass the “Comments to the Author” section, enter your conflict of interest statement in the “Confidential to Editor” section, and submit your "Accept" recommendation.

Reviewer #1: (No Response)

2. Is the manuscript technically sound, and do the data support the conclusions?

Reviewer #1: Yes

3. Has the statistical analysis been performed appropriately and rigorously? 

Reviewer #1: Yes

4. Have the authors made all data underlying the findings in their manuscript fully available?

Reviewer #1: Yes

5. Is the manuscript presented in an intelligible fashion and written in standard English?

Reviewer #1: No

6. Review Comments to the Author

Reviewer #1: I think the use of Fisher Z transforms on your data would be necessary for parametric tests, and your argument about the distributions of i th and jth correlation pairings potentially having different distributions is not particularly good, and only meaningful insofar as your number of subjects is too small. The correlations have different normal distributions, as you correctly state, but the central limit theorem indicates a sum of normal distributions will tend to normal. The distribution problems you may have with normality in your study are primarily due to the very small subject sample size you have (2 subjects) meaning the distribution might be multimodal if variances are low for different correlations across your analysis runs, because not enough subject variations for distributions are contributing. It would be better to directly acknowledge this as the basis of the non parametric testing. Since you are choosing using non-parametric tests here, rather than parametric tests, the ordinal testing you used will be unaffected by the z transform and it is not needed. The best study would have 6-10 subjects per group and use z-transformed correlation data.

I suggest a note to this effect, but it is your choice.

7. PLOS authors have the option to publish the peer review history of their article (what does this mean?). If published, this will include your full peer review and any attached files.

Reviewer #1: No

---

## [Editor Report · Acceptance letter]

17 Jun 2022

PONE-D-21-19322R2 

Comparing the reliability of different ICA algorithms for fMRI analysis 

Dear Dr. Wei:

I'm pleased to inform you that your manuscript has been deemed suitable for publication in PLOS ONE. Congratulations! Your manuscript is now with our production department. 

Kind regards, 

on behalf of

Dr. Pew-Thian Yap 

Academic Editor

PLOS ONE